# Increased Vascular Adhesion Protein 1 (VAP-1) Levels Are Associated with Alternative M2 Macrophage Activation and Poor Prognosis for Human Gliomas

**DOI:** 10.3390/diagnostics10050256

**Published:** 2020-04-27

**Authors:** Shu-Jyuan Chang, Hung-Pin Tu, Yen-Chang Clark Lai, Chi-Wen Luo, Takahide Nejo, Shota Tanaka, Chee-Yin Chai, Aij-Lie Kwan

**Affiliations:** 1Graduate Institute of Medicine, College of Medicine, Kaohsiung Medical University, Kaohsiung 80708, Taiwan; u100800001@kmu.edu.tw; 2Department of Public Health and Environmental Medicine, School of Medicine, College of Medicine, Kaohsiung Medical University, Kaohsiung 80708, Taiwan; p915013@kmu.edu.tw; 3Department of Pathology, Kaohsiung Medical University Chung Ho Memorial Hospital, Kaohsiung 80756, Taiwan; yichipan@hotmail.com; 4Division of Breast Surgery, Department of Surgery, Kaohsiung Medical University Chung Ho Memorial Hospital, Kaohsiung 80756, Taiwan; 1070598@kmuh.org.tw; 5Department of Surgery, Kaohsiung Medical University Chung Ho Memorial Hospital, Kaohsiung 80756, Taiwan; 6Department of Neurosurgery, Graduate School of Medicine, University of Tokyo, Tokyo 113-0033, Japan; tnejo-tky@umin.ac.jp (T.N.); tanakas-tky@umin.ac.jp (S.T.); 7Department of Pathology, College of Medicine, Kaohsiung Medical University, Kaohsiung 80708, Taiwan; 8Institute of Biomedical Sciences, National Sun Yat-Sen University, Kaohsiung 80424, Taiwan; 9Department of Neurosurgery, Kaohsiung Medical University Chung Ho Memorial Hospital, Kaohsiung 80756, Taiwan; 10Department of Surgery, Faculty of Medicine, College of Medicine, Kaohsiung Medical University, Kaohsiung 80708, Taiwan

**Keywords:** vascular adhesion protein 1, M2 macrophage activation, human gliomas, poor prognosis

## Abstract

Glioma is characterized by a high heterogeneity in the brain tumor. Abundant tumor-associated macrophages (TAMs) exist as neoplastic tissues, implicating tumor plasticity and thus leading to therapeutic challenges. Vascular adhesion protein (VAP-1) potentially serves as a mediator for TAM immunity in tumor milieu. We previously demonstrated that VAP-1 could contribute to tumor malignancy, but its characteristics in TAM immunity of glioma progression are still unclear. This study explored the association of VAP-1 expression with TAM distribution as well as the resulting clinical significance and prognostic value in human gliomas. An in-depth analysis of *AOC3* (VAP-1) gene expression was performed using 695 glioma samples derived from the cancer genome atlas (TCGA)-lower grade glioma and glioblastoma (GBMLGG) cohort. Bioinformatic analysis confirmed that VAP-1 expression is associated with poor prognosis of glioma patients (*p* = 0.0283). VAP-1 and TAM biomarkers (CD68, iNOS, and CD163) were evaluated by immunohistochemistry in 108 gliomas from Kaohsiung Medical University Hospital. VAP-1+ was expressed in 56 (51.85%) cases and this phenotype revealed a significant association with overall survival in Kaplan–Meier analysis (*p* < 0.0001). Immunohistochemical double staining showed that VAP-1 immunoreactivity was present around CD163+ M2 infiltration location, including aggressive lesions and neighboring neovasculature. We demonstrated that high VAP-1 expression levels positively correlated with CD163+ M2 activation and coexpression of these two proteins was associated with worse survival in gliomas (*p* < 0.0001). Multivariate analysis indicated that VAP-1 alone and co-expressed with CD163 were the significantly independent indicators (both *p* < 0.0001). Furthermore, VAP-1/CD163 coexpression exhibited excellent diagnostic accuracy in gliomas (AUC = 0.8008). In conclusion, VAP-1 and TAM CD163 M2 coexpression was found in glioma tissues belonging to a highly malignant subgroup that was associated with poor prognosis. These results implied VAP-1 abundance is closely linked to alternative M2 activation during glioma progression. From the aforementioned data, a reasonable inference is that VAP-1 combined with targeting M2 immunity might be an effective therapeutic target for human gliomas.

## 1. Introduction

Gliomas are the most common type of brain tumors, which are classified by the World Health Organization (WHO) into four grades according to the degree of malignancy [1]. Pilocytic astrocytoma (grade I) occurs more generally in younger people (less than 25 years), and it is defined as a non-malignant tumor with benign tendency. Diffuse astrocytic and oligodendroglial tumors are classified as grade II tumors, and belong to a slow-growing type of glioma in adults. Anaplastic astrocytoma and anaplastic oligoastrocytoma (grade III) are shown as infiltrative gliomas, and they display moderate pleomorphism and upraised mitotic activity but lack of microvascular proliferation or necrosis. Glioblastoma multiforme (GBM, grade IV), the most malignant gliomas, are characterized by undifferentiated, often pleomorphic cells with marked nuclear atypia and spirited mitotic activity. Rapid-growing, prominent microvascular proliferation and necrosis are the hallmark features in clinically aggressive GBMs that carry a worse prognosis [1]. The main roadblocks of malignant glioma treatment are the aggressive features, operational disadvantages, high recurrence rate, and therapeutic resistance.

An underlying pivotal factor in glioma progression is its intratumor heterogeneity; low tumor purity has been observed in malignant gliomas and it independently correlates with poor survival [2]. Aggressive gliomas particularly contain abundant non-neoplastic cells within the microenvironment, contributing to tumor adaptability and therapeutic challenges [2,3,4]. Tumor-associated macrophages (TAMs), the main population of non-neoplastic cells, are contingent upon activation of resident microglia and macrophages capable of immune modulation in brain tumors [4,5,6]. Activated TAMs are classified into two core phenotypes: classically activated macrophages (M1), and alternatively activated macrophages (M2) [7,8,9,10]. M1 induces Th1 proinflammatory responses that are involved in attacking tumor cells and normalizing aberrant neovascular networks to increase therapeutic sensitivity [11,12]. M1 macrophages induce inducible nitric oxide synthase (iNOS) that catalyzes the oxidation of arginine to produce a great deal of nitric oxide (NO) as a defense mechanism. NO causes not only cytotoxicity but also generation of numerous toxic metabolites that compose the M1 killing machinery. iNOS has been reported to increase M1 polarization and acts as a signature marker for M1-mediated immune responses [13,14]. Conversely, M2 engages in the Th2 immune response that encourages tumor progression through exaltation of angiogenesis, microenvironment remodeling, ectopic growth, and immunosuppression [8,9,15]. The majority of TAM subpopulations in tumor milieus exhibit an M2-like phenotype exerting immune tolerance and protumor effects [16,17]. Overall TAM subtypes were identified by CD68, a macrophage lineage marker [16,18]. M2 TAMs are specifically characterized with enhanced CD163 expression, attributable to the high expression of macrophage scavenger receptors by the M2 TAMs [17,19,20,21,22]. 

Studies reveal that the activation and recruitment of TAMs are triggered by a number of endothelial adhesion molecules [23,24], including vascular adhesion protein-1 (VAP-1). VAP-1 participates in the inflammatory regulation of the central nervous system and is involved in multiple inflammatory lesions and cerebrovascular diseases [25,26,27]. Also officially termed as amine oxidase copper containing three units (AOC3), VAP-1 is encoded by the *AOC3* gene on chromosome 17. This ectoenzyme is classified as the semicarbazide-sensitive amine oxidase (SSAO, EC.1.4.3.21) that oxidizes primary amines in a reaction producing hydrogen peroxide, aldehyde, and ammonia. It also serves as a multifunctional molecule existing in pericytes and vascular endothelium and is chiefly engaged in leukocyte tethering and trafficking to inflamed tissues under physiological conditions [28,29,30]. 

Some investigators have shown that VAP-1 is necessary to accelerate neoangiogenesis and tumor growth via enhancing the recruitment of myeloid-derived suppressor cells into tumors [31,32]. Further studies have revealed that VAP-1 functions as an endothelial activation marker that is elicited when metastatic tumor cells are attached to the vascular bed, leading to TAM recruitment and metastatic cell survival [33,34,35]. Notably, VAP-1 encourages IL-1β–stimulated M2 macrophage recruitment and infiltration that contribute to lymphogenesis and angiogenesis [36]. Recent works have revealed that VAP-1 is strongly expressed in angiogenic diseases and malignant neoplasms [37,38,39,40], and its gene amplification has been verified in the genome of cancer [41]. We previously reported that increased VAP-1 expression correlated with advancing grades and worse outcome in astrocytoma patients [37]. However, the effects of VAP-1 in TAM immunity during glioma progression are still uncertain. The purpose of the current study was to investigate the relationship between altered VAP-1 expression and TAM distribution as well as prognosis in human gliomas.

## 2. Materials and Methods 

### 2.1. AOC3 Exon Expression and DNA Methylation Datasets

*AOC3* gene expression in the prognosis evaluation in human glioma was illustrated from the bioinformation analysis of TCGA lower grade glioma and glioblastoma (GBMLGG) cohort. This cohort was composed of brain low-grade glioma (LGG) and glioblastoma (GBM). The LGG group consisted of patients with astrocytomas, oligodendrogliomas, and oligoastrocytomas, while glioblastoma multiforme was included in the GBM group. This cohort contained twenty-seven recurrent cases. Level 3 data, the calculated expression signal of a particular composite exon of a gene, were downloaded using the University of California Santa Cruz (UCSC) Xena browser (https://xenabrowser.net). Glioma samples without exon sequencing or methylation data were excluded. After screening with criteria, 695 glioma tissues were identified as eligible samples for *AOC3* gene expression, whereas 681 available samples were included in *AOC3* methylation analysis. 

*AOC3* transcriptional profile was identified experimentally using the Illumina HiSeq 2000 RNA Sequencing platform by the University of North Carolina TCGA genome characterization center. Level 3 data for each sample, was downloaded from the TCGA data coordination center (DCC). This dataset contained three exons: chr17:41003201-41004960, chr17:41006465-41006750, and chr17:41007461-41007590, and it displayed the exon level transcription evaluations as in RPKM (reads per kilobase of exon model per million mapped reads) values. Exons were mapped onto the human genome coordinates using the UCSC Xena unc_RNAseq_exon probe Map and referenced to a method description from the University of North Carolina TCGA genome characterization center: DCC description. The RPKM values of three exons from *AOC3* gene were averaged and reported as the *AOC3* gene expression. 

*AOC3* DNA methylation profile was measured experimentally using the Illumina Infinium Human Methylation 450 platform. DNA methylation beta values were recorded for each array probe in each sample via Bead Studio software, which were derived at the Johns Hopkins University and University of Southern California TCGA genome characterization center. This dataset involves twelve methylation probes, namely, cg22530519, cg16048817, cg24662231, cg09040752, cg08834922, cg21602160, cg11744144, cg25512683, cg16066544, cg19055390, cg21308545, and cg08562004, and a bimodal distribution of the beta value was observed. *AOC3* methylation beta values are continuous variables between 0 and 1, which represent the ratio of the intensity of the methylated bead type to the combined locus intensity. In general, high beta values indicated hypermethylation and lower beta values represented hypomethylation. Microarray probes were mapped onto the human genome coordinates using the UCSC Xena probe Map derived from GEO GPL13534 record and referenced to Illumina Infinium Bead Chip DNA methylation platform beta values. The beta values of twelve methylation probes from *AOC3* gene were averaged to yield the *AOC3* methylation status.

### 2.2. Specimens

Tumor samples were retrieved from 108 patients with resected glioma as a part of routine clinical care at the Cancer Center of Kaohsiung Medical University Hospital (KMUH) over the period of 2008–2016 (Table 1). All patients were carefully screened for the presence of malignancies by expert pathologists, and the gliomas were classified according to the WHO 2016 criteria [1]. All tumor tissues included in this study were from patients with complete clinical follow-up information. The clinicopathological information was obtained from the cancer center and collected from the medical records of patients, including age, gender, WHO grade, tumor size, and recurrence status. The overall survival time was calculated from the date of first diagnosis until the last follow-up or until death of the patients, with a maximum follow-up time of 60 months. Patients who had insufficient clinical and/or pathology information were excluded from this study. Approval to link laboratory data to clinical and pathologic data was obtained from the Institutional Review Board (KMUHIRB-E(I)20180113).

### 2.3. Immunohistochemistry (IHC)

Paraffin-embedded sections (3 μm thick) were de-paraffined in xylene and dehydrated through graded alcohols. Antigen retrieval was carried out with Dako Target Retrieval Solution, Citrate pH6 (S236984, DAKO, Glostrup, Denmark) at 121 °C for 10 min. The slides were then incubated in 3% hydrogen peroxide at room temperature to quench endogenous peroxidase. Anti-human mouse monoclonal antibodies were used to detect VAP-1 (sc-166713, Santa Cruz Biotechnology, Dallas, TX, USA), CD68 (M0876, DAKO, Glostrup, Denmark), and CD163 (NCL-L-CD163, Leica Biosystems, Wetzlar, Germany), whereas anti-human rabbit polyclonal antibodies and rabbit monoclonal antibodies were utilized to identify iNOS (SAB4502012, Sigma Aldrich, St. Louis, MO, USA) and CD163 (ab182422, Abcam, Cambridge, MA, USA), respectively. IDH1 mutation status was detected by IDH1R132H antibody (DIA-H09, Dianova GmbH, Hamburg, Germany). Incubation with primary antibodies was performed overnight at 4 °C. The antigen-antibody complexes were visualized using the DAKO REAL Envision Detection System, Peroxidase/DAB, Rabbit/Mouse (K5007, DAKO, Glostrup, Denmark), followed by hematoxylin counterstaining and mounting. DoubleStain IHC Kit: Mouse & Rabbit on human tissue (DAB & AP/Red) (ab210059, Abcam, Cambridge, MA, USA) was used in immunohistochemistry to evaluate two distinct antigens in a single tissue.

### 2.4. Pathologic Evaluation

Two pathologists who were blinded to all clinical data independently evaluated all samples and scored for protein immunoreactivity as well as for intensity of positively stained tumor cells. Staining for each of the proteins was scored using the methods of modified immunoreactive score (IRS) [42,43]. Staining for each of the proteins in each slide was given a score of 0, 1, 2, or 3 if none, <10%, 10–50%, or >50%, respectively, of the cells were positively stained. The intensity score represented the staining intensity (0, no staining; 1, weak; 2, moderate; 3, strong staining). In the modified IRS method, the scores for the percentages of positively stained cells (0–3) and staining intensity (0–3) were multiplied to give a total score. The total score ranged from 0 to 9, with 4 or lower being defined as negative/low expression and 6 or greater as positive expression.

### 2.5. Statistical Analysis

The association between *AOC3* gene expression and clinical characteristics, such as WHO grades, recurrence, and patient survival in subgroups of low-grade glioma and glioblastoma was performed by box-whisker plots. The chi-square test was used to assess the differences between *AOC3* expression and clinicopathological parameters. Kaplan–Meier plots were utilized to analyze the association of *AOC3* expression and methylation status with overall survival in the TCGA glioma cohort. Descriptive statistics of the study population, including means (with corresponding standard deviations), medians (with corresponding ranges), and proportions were computed. Comparison of target protein expressions between glioma patients in different clinicopathological groups were evaluated by the chi-square test. The relationships between the expression patterns of VAP-1 and TAM subgroups were determined by Pearson’s correlation coefficient. Survival and hazard functions were estimated using the Kaplan–Meier method, and survival between groups was compared using the two-sided log-rank test (log-rank test for trend only when ≥three groups were entered in logical order). Cox proportional hazards regression model was used to identify risk factors related to survival after adjusting for other factors. The diagnostic accuracies of individual parameters were compared by using receiver operating characteristic curves. A *p*-value of < 0.05 was deemed to indicate statistical significance. Box and whisker plots and stacked bars were generated by using SigmaPlot 10.0 (Systat Software Inc., CA, US) and Prism software 5.0 (GraphPad Prism Software, La Jolla, CA, USA); Kaplan–Meier curves and statistical analyses were carried out using SAS 9.3 (SAS Institute Inc, NC, US).

## 3. Results

### 3.1. VAP-1(AOC3) Expression as a Potential Biomarker for Prognosis in Patients with Gliomas

Publicly available large-scale database and clinical parameters derived from TCGA lower grade glioma and glioblastoma (GBMLGG) cohort were used to assess *AOC3* gene expression in different grades of gliomas. The potential value of *AOC3* gene expression in the prognosis evaluation in gliomas was validated by 529 LGG cases and 166 GB cases available from the exon expression and by the DNA methylation datasets. We preliminarily investigated *AOC3* gene expression and methylation status in the glioma subgroups and downloaded the level 3 data from UCSC Xena (Figure 1A). Analyses of *AOC3* exon expression and DNA methylation in glioma subgroups were conducted using UCSC Xena (Figure 1B,C). Three exons were used in *AOC3* transcriptional dataset, and twelve methylation probes were utilized in the gene methylation profile study as described in the Section 2. *AOC3* exhibited greater expression in the GBM group compared with that of the LGG group (chr17:41007461-41007590:+, *p* = 0.0024, Figure 1B). In contrast with exon expression, lower methylation statuses of *AOC3* were observed in the GBM patients (all methylation probes, *p* < 0.0001; Figure 1C). Further, the association between *AOC3* gene expression and clinical characteristics in glioma subgroups was observed by chi-square test and performed by box and whisker plots. The patients were divided into low (−) and high (+) expression groups based on the cut-off of the median of the *AOC3* exon expression (0.4471) and methylation status (0.7207). *AOC3* gene expression positively correlated with high WHO grade, recurrence, and worse survival in glioma subgroups (*p* = 0.0081, *p* = 0.0117, and *p* = 0.0216 respectively; Figure 1D). However, lower DNA methylation status of *AOC3* showed worse prognosis for glioma patients (all *p* < 0.0001; Figure 1E). *AOC3* as a valid prognostic marker for human glioma was displayed by Kaplan–Meier plots (exon, *p* = 0.0283; methylation, *p* < 0.0001; Figure 1F–G). *AOC3* gene expression analysis was consistent with our previous findings [37]. Thus, VAP-1 can be used as a prognostic marker in patients with gliomas. 

### 3.2. VAP-1 Expression and Co-Expressed with TAM Biomarkers are Elevated in Malignant Glioma Tissues 

In the tumor microenvironment, VAP-1 is capable of exacerbating tumor malignancy through enhancing leukocyte recruitment and contributing to TAM activation [31,32,33,36,44]. Therefore, it is important to identify the relationship between VAP-1 and TAM properties in glioma progression. One hundred and eight glioma patients were enrolled in this study; descriptive statistics of the study population are shown in Table 1. VAP-1 and TAM biomarkers in glioma tissues were evaluated by immunohistochemistry, and the representative images of all proteins are shown in Figure 2A–D. The populations of overall TAMs, M1, and M2 macrophages in neoplastic lesions were measured with anti-CD68, anti-iNOS, and anti-CD163 antibodies, respectively [4,45]. As described in the Section 2, the scoring system used for evaluating the expression of VAP-1 protein consisted of both the extent of tumor cells and the intensity of immunopositivity in the cytoplasm. Expressions of CD68, iNOS, and CD163 in the nuclear and cytoplasm were scored using the same method. 

Relatively prominent expression of VAP-1 was found in the malignant gliomas and the surrounding neovasculature (Figure 2A, bottom panel). A significant increase in TAM immunoreactivity of glioma tissues was observed in the apparently aggressive areas (Figure 2B–D, bottom panel). From double-staining, it was found that iNOS+ M1 and CD163+ M2 existed in different glioma areas (Figure 2E,F). In these figures, VAP-1 was visualized with red color (cytoplasmic staining), whereas iNOS and CD163 with brown color (nuclear pattern). Glioma tissues infiltrated by iNOS+ M1 macrophages were shown to have a low level or be absent of VAP-1 expression (Figure 2E, left figure). In contrast, VAP-1 expressed in neighboring locations with CD163+ M2 macrophage was identified from tumor milieu (Figure 2E,F). In addition, correlation between VAP-1 expression and TAM biomarkers were analyzed in glioma patients. Immunoreactivity percentage of VAP-1 co-expressed with CD68, iNOS, and CD163 phenotypes were observed in lesions (Figure 2G–I). High percentages of glioma patients with positive VAP-1/CD68 (65.75%) and VAP-1/CD163 (73.13%) co-immunoreactivity were apparently observed (Figure 2H,I). These results showed that the immunoreactivity of VAP-1 co-localized not only with total TAM but also with M2 in glioma tissues.

### 3.3. VAP-1 Alone and VAP-1/TAM Coexpression Correlated with Clinicopathological Variables in Glioma Patients

The patient demographic, clinical, and pathological characteristics are shown in Table 2. High VAP-1 expression was detected in 56 (51.85%) out of 108 patients, and it was strongly correlated with advanced WHO grades (*p* < 0.0001), poor survival (*p* < 0.0001) and IDH1 mutant (*p* = 0.0262, Table 2). The tumor tissues presenting VAP-1/CD68, VAP-1/iNOS, and VAP-1/CD163 coexpression were detected in 47 (43.52%), 29 (26.85%), and 49 (45.37%) specimens respectively (Table 2). VAP-1/CD68+ TAM was significantly increased with WHO grade (*p* < 0.0001) and poor survival (*p* < 0.0001, Table 2). There was positive association with VAP-1+/iNOS+ M1 phenotype and pathological grade (*p* < 0.0001) and patient survival (*p* = 0.0165, Table 2). A significant increasing trend in the VAP-1+/CD163+ M2 phenotype was observed in cancer specimens for age (*p* = 0.0164), WHO grade (*p* < 0.0001), patient survival (*p* < 0.0001) and IDH mutations (*p* = 0.0053, Table 2). Besides, the results of correlation coefficients confirmed VAP-1 expression was significantly correlated with distribution of CD68+ TAMs and CD163+ M2 macrophages (both *p* < 0.0001), but not with that of iNOS+ M1 (Table 3).

### 3.4. Impact of VAP-1 and VAP-1/TAM Phenotype on the Survival of Glioma Patients

The prognostic effects of VAP-1 alone or VAP-1/TAM coexpression on overall survival (OS) was assessed by Kaplan–Meier plots. Overall survival was significantly lower in patients with positive VAP-1 and CD163 phenotypes (both *p* < 0.0001, Figure 3A and Figure A1), but not in patients with positive CD68 or iNOS immunoreactivities (*p* = 0.0509 and *p* = 0.1174, Figure A1). Coexpression of VAP-1 and CD68+ total TAMs, iNOS+ M1, CD163+ M2 displayed significant prognostic effects on patient survival (*p* < 0.0001, *p* = 0.0101 and *p* < 0.0001, respectively, log-rank test; Figure 3B–D).

Table 4 shows the univariate analysis results for the glioma patients from KMUH. The results showed that VAP-1 expression (*p* < 0.0001), age (*p* = 0.0129), and mutant-IDH1 status (*p* = 0.0011) were significantly associated with reduced overall survial. Moreover, it also demonstrated that positive VAP-1/TAM markers co-immunoreactivities yielded poorer prognosis for glioma patients (VAP-1/CD68, *p* < 0.0001; VAP-1/iNOS, *p* = 0.0113; VAP-1/CD163, *p* < 0.0001; Table 4). In multivariate regression analysis, reference parameters, such as gender, age, tumor size, recurrence, and IDH1 mutation were used. VAP-1 expression displayed as an independent factor on patient outcome (HR: 4.688, 95% confidence interval (CI): 2.736–8.033; *p* < 0.0001; Table 4). Multivariate analysis adjusted by gender, age, tumor size, recurrence, and IDH1 mutant status indicated the hazard ratio for VAP-1 co-expressed with CD68 or CD163 enhanced the prognostic capability significantly in gliomas (VAP-1/CD68, HR: 3.226, 95% CI: 1.980–5.256, *p* < 0.0001; VAP-1/CD163, HR: 6.597, 95% CI: 3.677–11.836, *p* < 0.0001; Table 4). It appeared VAP-1/iNOS coexpression could serve as a valuable biomarker for patient survival (univariate, *p* = 0.0113), but this phenotype had no significant influence in multiple logistic regression models (*p* = 0.0817, Table 4).

### 3.5. Diagnostic Accuracies of VAP-1 Expression and VAP-1/TAM Coexpression

Receiver operating characteristic (ROC) curves were utilized to determine whether VAP-1 alone and VAP-1/TAM coexpression were potential specific biomarkers for glioma diagnosis (Figure 4). The area under ROC curve (AUC) of VAP-1 was 0.7730, showing that VAP-1 expression appeared to be a good discriminating marker for gliomas (Figure 4A). CD163 expression alone is also a potential biomarker for diagnosis of glioma patients (AUC = 0.7565, Figure A2 ), whereas neither CD68 nor iNOS expression exhibits the diagnostic accuracy (CD68, AUC = 0.5851; iNOS, AUC = 0.6013) (Figure A2). VAP-1/CD68 denoted acceptable accuracy for glioma prognosis (AUC = 0.7300, Figure 4B). Increased AUC of VAP-1/CD163 coexpression further demonstrated better predictive accuracy for glioma than did VAP-1 expression alone (AUC = 0.8008, Figure 4D).

## 4. Discussion

This study is the first to report that VAP-1 either alone or co-expressed with M2 immunity could be a prognostic marker of disease progression and patient outcome in human gliomas. Malignant gliomas and metastatic patterns have highly aggressive and heterogeneous features causing resistance to therapies [4,7]. TAMs are the major population of non-neoplastic cells surrounding or within glioma tissues, and they contribute to tumor plasticity and treatment failure [45,46]. VAP-1 is regarded as one of the critical components for tumor progression by altering immunomodulatory properties, especially mediating TAM infiltration as well as alternative M2 activation within the tumor microenvironment [31,32,33,36,44]. We previously demonstrated that a positive correlation between VAP-1 pattern and astrocytoma malignancy is conducive to outline the disease progression [37]. 

In this study, a powerful confirmation of the prognostic potential of *AOC3* gene expression was illustrated from the 695 glioma samples that were derived from TCGA-LGGGBM cohort. *AOC3* gene expression positively correlated with grade, recurrence, and patient survival in gliomas (Figure 2B), but the DNA methylation status was inversely related to disease progression and outcomes in this tumor. Low levels of DNA methylation occurred in worse prognostic glioma, which might imply demethylation as an epigenetic modification to enhance gene expression in accordance with high levels of *AOC3* gene expression in patients with poor outcome. Excessive VAP-1 presented in malignant tumors is reported as having positive association with neoplastic disorders as also seen in the authors’ work [31,32,33,36,38,39,40,41,44]. In this study, 108 glioma patients were evaluated in this study with 56 patients presenting positive VAP-1 phenotype, which closely paralleled earlier reports that malignant neoplastic tissues expressed higher VAP-1 immunohistochemically [37]. 

Although VAP-1 has been considered as a biomarker for some cancer types, the effect of VAP-1-associated TAM immunity upon glioma progression is still unclear. The aims of this study were to determine whether or not in situ expression of VAP-1 and its coexpression with TAM populations presented prognostic value in human gliomas. We addressed total TAMs and two main TAM subtypes in glioma tissues. CD68 is an indicator of total TAM population, while iNOS and CD163 are the major identifiers of M1 and M2 macrophages, respectively. An elevated number of TAMs found along and inside the lesions and clustered around neovessels or necroses was shown to closely correlate with glioma maintenance and progression [4,46,47,48,49,50]. Consistent with these studies, the present study also demonstrated the proportions of TAMs greatly outnumbered other non-neoplastic subgroups in glioma milieu and were progressively correlated with the glioma grades (Table 2 and Figure 2). On the basis of the VAP-1+/CD163+ staining results in double-immunostained sections, activated M2 macrophages were found to be clustered around the VAP-1 protein (Figure 2E–F). It’s worth noting that VAP-1 and M2 subtype, but not M1 subtype, existed in adjacent locations of glioma tissues, especially around the neovasculature and near aggressive regions. 

In a total of 108 glioma patients, VAP-1 expression and VAP-1/CD163 coexpression were both correlated with age, grade, survival, and IDH1 mutations. Crucially, they were also associated with a high risk of worse outcome in glioma patients. This study revealed that VAP-1 and VAP-1/CD163 phenotypes are both independent prognostic indicators of worst prognosis. In particular, VAP-1/CD163 coexpression showed excellent diagnostic accuracy in gliomas (AUC = 0.8008). Although VAP-1/CD68 and VAP-1/iNOS also exhibited their diagnostic properties for overall survival, they did not exhibit the same degree of diagnostic accuracy in gliomas as that of VAP-1/CD163 (VAP-1/CD68, AUC = 0.7300; VAP-1/iNOS, AUC = 0.5535) (Figure 4). 

In summary, the clinicopathologic values of VAP-1 alone and in combination with TAM M2 subtype for predicting the prognosis of gliomas are reported. These data are consistent with recent studies suggesting that VAP-1 modulates M2 macrophage infiltration, which immediately causes lymphangiogenesis and angiogenesis to exacerbate cancer malignancy [36]. Several studies demonstrated that M2 TAMs display proangiogenic properties by secreting a variety of angiogenic factors and cytokines, while engaging in a number of heterotypic interactions [51,52,53,54,55]. VAP-1 selectively enhances M2-like TAM recruitment depending on the implanted cytokines [24,31], and might thus indirectly contribute to M2 macrophage-mediated angiogenesis in tumors [36]. Therefore, a reasonable inference is that attenuating VAP-1-mediated M2 properties might be an effective therapeutic strategy for the treatment of malignant gliomas. Our future work will focus on elucidating detailed mechanisms by which VAP-1 influences the M2 immunity in glioma under in vitro and in vivo conditions. 

## 5. Conclusions

In conclusion, elevated levels of VAP-1 protein in gliomas constitute a highly aggressive subgroup associated with poor outcomes, and VAP-1 co-expressed with TAM CD163 subtype is shown to be an excellent prognostic indicator for glioma patients. Overall, VAP-1 abundancy is strongly linked to alternative M2 activation that might contribute to tumor immunity during glioma progression. All these results provide an important stepping stone for future studies and offer new tactics for glioma therapy.

## Figures and Tables

**Figure 1 diagnostics-10-00256-f001:**
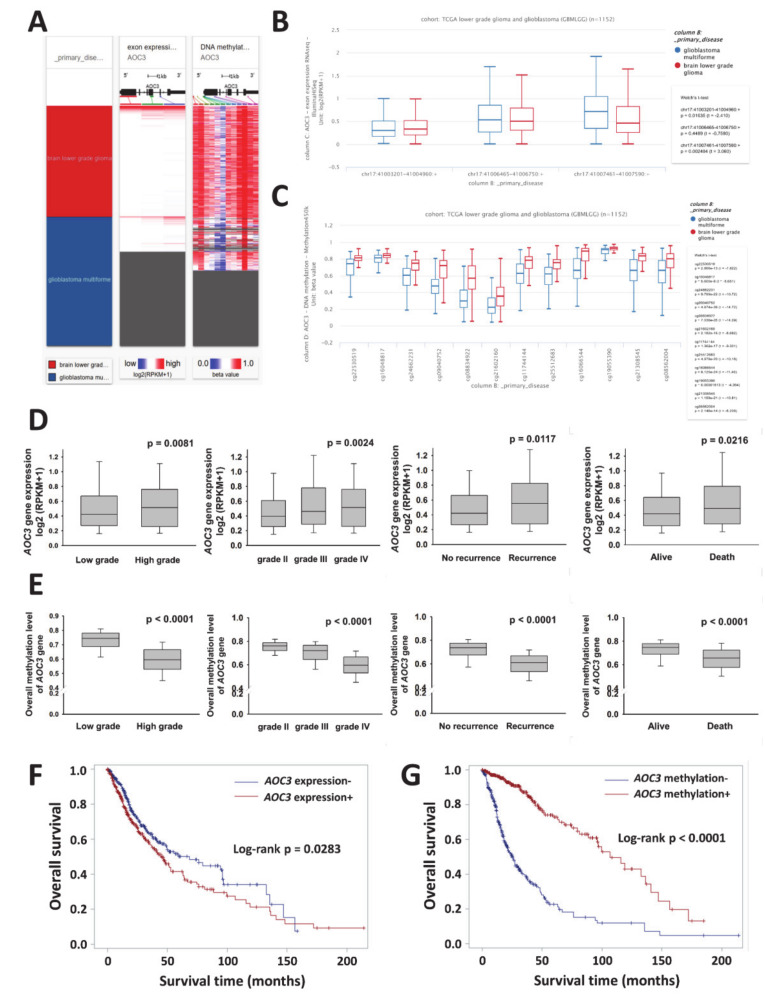
Identification of the predicted potential of *AOC3* in glioma from TCGA low-grade glioma and glioblastoma dataset. (**A**) *AOC3* gene expression and methylation status in the glioma subgroups were analyzed and the level 3 data were downloaded from UCSC Xena. Three exons were used in *AOC3* transcriptional dataset, and twelve methylation probes were utilized in the gene methylation profile study as described in the Section 2. Analyses of *AOC3* exon expression (**B**) and DNA methylation (**C**) in subgroups of low-grade glioma and glioblastoma were performed using UCSC Xena. The levels of *AOC3* exon expression (**D**) and methylation status (**E**) in different subgroups of gliomas were shown by box and whisker plots. Kaplan–Meier curves were performed with exon expression (**F**) and DNA methylation (**G**) datasets and showed association of *AOC3* gene expression with overall survival in glioma patients. *p* < 0.05 was considered statistically significant. Abbreviations: TCGA: the cancer genome atlas; UCSC: University of California Santa Cruz.

**Figure 2 diagnostics-10-00256-f002:**
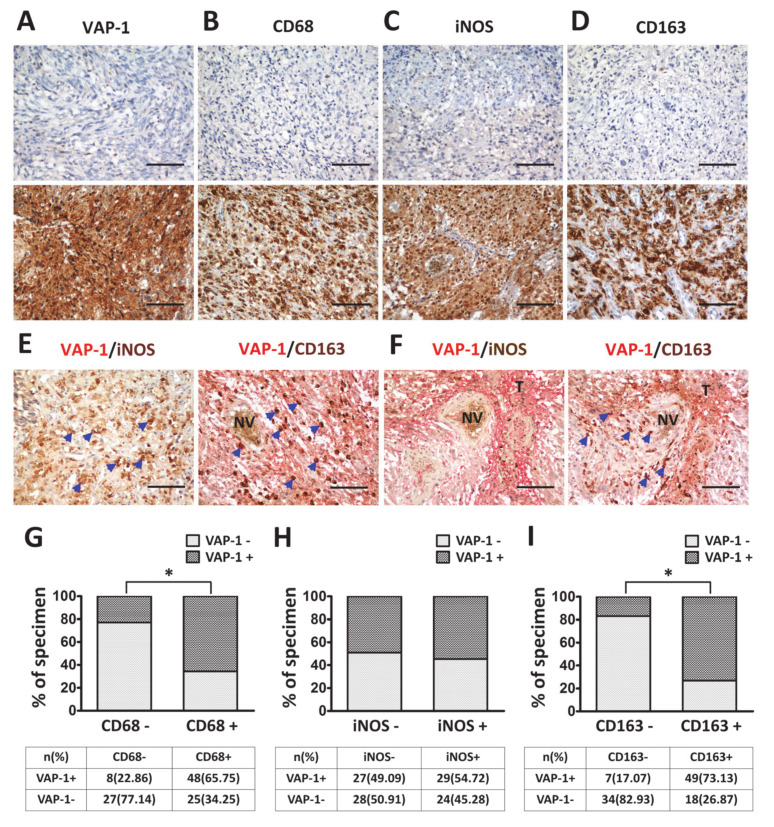
Representative immunostaining for expressions of VAP-1 and TAM-related proteins in gliomas. (**A**) Immunoreactivity of VAP-1: Top panel, negative in tumor cells as determined from staining in the cytoplasm; Bottom panel, positive staining. Immunoreactivity of TAM identifier, CD68 (**B**), iNOS (**C**), and CD163 (**D**) revealed nuclear and cytoplasmic staining in gliomas, which indicated tumor immune activity. (**E**, **F**) Double-staining immunohistochemistry of VAP-1(red color, cytoplasmic staining) in combination with iNOS+ M1 or CD163+ M2 (brown color, nuclear staining) showed two TAM subtypes presented at different levels of VAP-1 from glioma milieu. Blue arrows indicate the iNOS+ or CD163+ macrophages in tumor tissues. Immunoreactivity percentage of VAP-1 co-expressed with CD68 (**G**), iNOS (**H**), and CD163 (**I**) phenotypes were observed in lesions. Low VAP-1 expression is depicted as light gray columns, whereas high VAP-1 expression is shown as dark gray columns. Chi-square test was used for statistical analysis. (Original magnification: ×200). Abbreviations: VAP-1: vascular adhesion protein-1; TAM: tumor associated macrophage; iNOS: inducible nitric oxide synthase; T, tumor tissue; NV, neovasculature. *p* < 0.05 was considered statistically significant.

**Figure 3 diagnostics-10-00256-f003:**
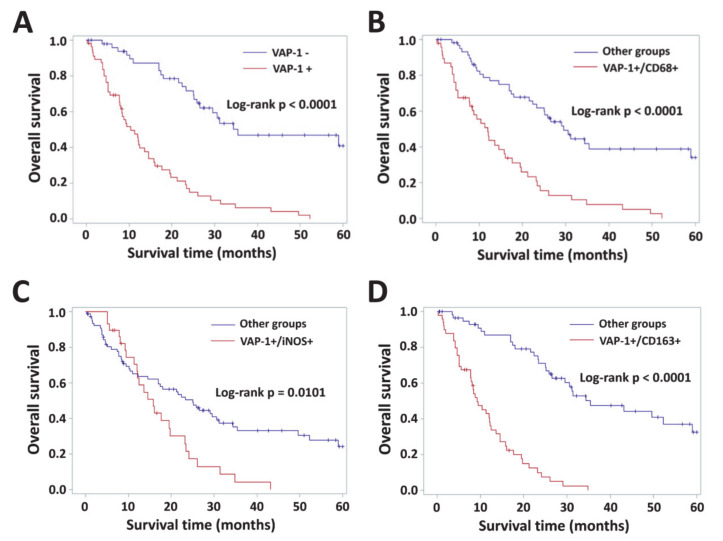
Kaplan–Meier plots for glioma patients with different levels of VAP-1 alone or VAP1/TAM coexpression. (**A**) VAP-1 expression had a significant effect on survival in 108 patients with glioma. Survival in patients that had tumors with positive coexpression of VAP-1/TAM markers (**B**, CD68; **C**, iNOS; **D**, CD163) in comparison with other phenotypes was observed. *p* < 0.05 was considered statistically significant.

**Figure 4 diagnostics-10-00256-f004:**
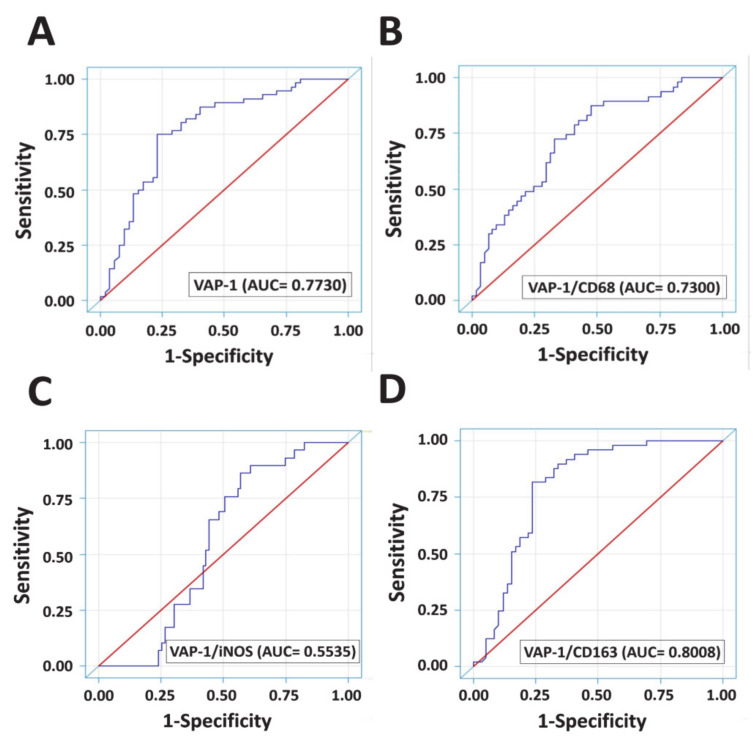
Analysis for the diagnostic accuracy of using VAP-1 expression and VAP-1/TAM coexpression in gliomas by ROC curve. ROC analyses for the diagnostic accuracy of using VAP-1 expression alone (**A**) and in combination with TAM markers (**B**, CD68; **C**, iNOS; **D**, CD163) were calculated. The AUC estimates evaluated by this approach are reported under the ROC association statistics section of data output. Abbreviations: AUC, area under the ROC curve. Abbreviations: ROC curves, receiver operating characteristic curves; AUC, the area under ROC curve.

**Table 1 diagnostics-10-00256-t001:** Patient baseline characteristics (*n* = 108).

Characteristics	Median (Range) or *n* (%)
Total number of patients	108
Age (years) mean ± SD medians (corresponding ranges)	50.26 ± 17.51 52.00 (20.00–83.00)
Tumor size (cm) mean ± SD medians (corresponding ranges)	1.88 ± 1.32 1.60(0.20–7.70)
Follow-up of the patient cohort (months) mean ± SD medians (corresponding ranges)	20.88 ± 17.39 16.82 (0.23–60.00)
Gender, *n* (%) Female Male	40 (37.04) 68 (62.96)
WHO grade, *n* (%) II III IV	27 (25.00) 35 (32.41) 46 (42.59)
Histological type, *n* (%) diffuse astrocytoma oligoastrocytoma oligodendroglioma anaplstic astrocytoma GBM	28 (25.93) 10 (9.26) 3 (2.78) 28 (25.93) 39 (36.11)
Recurrence, *n* (%) Absent Present	53 (49.07) 55 (50.93)
Survival status, *n* (%) survived died	34 (31.48) 74 (68.52)
IDH1 mutant, *n* (%) Negative Positive	81 (75.00) 27 (25.00)

SD: standard deviation; WHO: World Health Organization; GBM: glioblastoma multiforme; IDH1: isocitrate dehydrogenase. Descriptive statistics of the study population, including means (with corresponding standard deviations), medians (with corresponding ranges), and proportions were computed.

**Table 2 diagnostics-10-00256-t002:** Associations between VAP-1 or combined VAP-1/TAM-related proteins and known clinicopathological parameters in 108 patients with glioma.

Parameters	VAP-1- *n* (%)	VAP-1+ *n* (%)	*p* Value	Other Groups * *n* (%)	VAP-1+/ CD68+ *n* (%)	*p* Value	Other Groups ** *n* (%)	VAP-1+/ iNOS+ *n* (%)	*p* Value	Other Groups *** *n* (%)	VAP-1+/ CD163+ *n* (%)	*p* Value
	52 (48.15)	56 (51.85)		61 (56.48)	47 (43.52)		79 (73.15)	29 (26.85)		59 (54.63)	49 (45.37)
**Gender** Female Male	21 (40.38) 31 (59.62)	19 (33.93) 37 (66.07)	0.4876	25 (40.98) 36 (59.02)	15(31.91) 32(68.09)	0.3333	31 (39.24) 48 (60.76)	9 (31.03) 20 (68.97)	0.4338	24 (40.68) 35 (59.32)	16 (32.65) 33 (67.35)	0.3899
**Age** ≤45 years >45 years	25 (48.08) 27 (51.92)	17 (30.36) 39 (69.64)	0.0591	28 (45.90) 33 (54.10)	14 (29.79) 33 (70.21)	0.0885	33 (41.77) 46 (58.23)	9 (31.03) 20 (68.97)	0.3104	29 (49.15) 30 (50.85)	13 (26.53) 36 (73.47)	0.0164
WHO grade II III IV	24 (46.15) 21 (40.38) 7 (13.46)	3 (5.36) 14 (25.00) 39 (69.64)	<0.0001	24 (39.34) 22 (36.07) 15 (24.59)	3 (6.38) 13 (27.66) 31 (65.96)	<0.0001	26 (32.91) 31 (39.24) 22 (27.85)	1 (3.45) 4 (13.79) 24 (82.76)	<0.0001	25 (42.37) 24 (40.68) 10 (16.95)	2 (4.08) 11 (22.45) 36 (73.47)	<0.0001
Tumor size <2 cm ≥2 cm	33 (63.46) 19 (36.54)	30 (53.57) 26 (46.43)	0.2976	39 (63.93) 22 (36.07)	24 (51.06) 23 (48.94)	0.1786	47 (59.49) 32 (40.51)	16 (55.17) 13 (44.83)	0.6864	38 (64.41) 21 (35.59)	25 (51.02) 24 (48.98)	0.1601
**Recurrence** Absent Present	25 (48.08) 27 (51.92)	28 (50.00) 28 (50.00)	0.8417	30 (49.18) 31 (50.82)	23 (48.94) 24 (51.06)	0.9799	41 (51.90) 38 (48.10)	12 (41.38) 17 (58.62)	0.3325	29 (49.15) 30 (50.85)	24 (48.98) 25 (51.02)	0.9857
Survival status survived died	29 (55.77) 23 (44.23)	5 (8.93) 51 (91.07)	<0.0001	29 (47.54) 32 (52.46)	5 (10.64) 42 (89.36)	<0.0001	30 (37.97) 49 (62.03)	4 (13.79) 25 (86.21)	0.0165	30 (50.85) 29 (49.15)	4 (8.16) 45 (91.84)	<0.0001
**IDH1 mutant** Negative Positive	34 (65.38) 18 (34.62)	47 (83.93) 9 (16.07)	0.0262	43 (70.49) 18 (29.51)	38 (80.85) 9 (19.15)	0.2177	57 (72.15) 22 (27.85)	24 (82.76) 5 (17.24)	0.2592	38 (64.41) 21 (35.59)	43 (87.76) 6 (12.24)	0.0053

WHO: World Health Organization; GBM: glioblastoma multiforme; IDH1: isocitrate dehydrogenase 1. Other groups * the groups excepting VAP-1+/CD68+ (include VAP-1-/CD68-, VAP-1-/CD68+, and VAP-1+/CD68- groups). Other groups ** the groups excepting VAP-1+/iNOS+ (include VAP-1-/iNOS-, VAP-1-/iNOS+, and VAP-1+/iNOS- groups); Other groups *** the groups excepting VAP-1+/CD163+ (include VAP-1-/CD163- , VAP-1-/CD163+, and VAP-1+/CD163- groups). Chi-square test was used for statistical analysis. *p* < 0.05 was considered statistically significant.

**Table 3 diagnostics-10-00256-t003:** Assessment of the relationship between VAP-1 and TAM associated markers (CD68, iNOS, and CD163) by Pearson’s correlation coefficient (*n* = 108).

Variable	VAP-1		CD68		iNOS			CD163	
	Correlation	*p* Value	Correlation	*p* Value		Correlation	*p* Value	Correlation	*p* Value
VAP-1	1.00000	–	0.40181	<0.0001		0.05629	0.5628	0.54450	<0.0001
CD68	0.40181	<0.0001	1.00000	–		0.04654	0.6325	0.35519	0.0002
iNOS	0.05629	0.5628	0.04654	0.6325		1.00000	–	0.11909	0.2196
CD163	0.54450	<0.0001	0.35519	0.0002		0.11909	0.2196	1.00000	–

The association between all proteins were subjected to Pearson’s correlation coefficient. *p* < 0.05 was considered statistically significant.

**Table 4 diagnostics-10-00256-t004:** Univariate and multivariate Cox proportional hazard models with backwards elimination for predicting tumor progression in glioma patients.

Parameters	Univariate		Multivariate							
	HR (95% CI)	*p*-Value	VAP-1	*p*-Value	VAP-1/CD68	*p*-Value	VAP-1/iNOS	*p*-Value	VAP-1/CD163	*p*-Value
			HR (95% CI)		HR (95% CI)		HR (95% CI)		HR (95% CI)	
VAP-1	5.057 (3.016–8.481)	<0.0001	4.688(2.736–8.033)	<0.0001	–	–	–	–	–	–
VAP-1/CD68	3.483 (2.164–5.605)	<0.0001	–	–	3.226 (1.980–5.256)	<0.0001	–	–	–	–
VAP-1/iNOS	1.908 (1.157–3.145)	0.0113	–	–	–	–	1.572 (0.945–2.615)	0.0817	–	–
VAP-1/CD163	7.047 (4.085–12.155)	<0.0001	–	–	–	–			6.597 (3.677–11.836)	<0.0001
Gender (female = 1)	0.956 (0.597–1.531)	0.8513	0.735 (0.426–1.267)	0.2675	0.725 (0.419–1.252)	0.2480	0.736 (0.436–1.240)	0.2495	0.652 (0.372–1.140)	0.1336
Age	1.853 (1.140–3.012)	0.0129	1.816 (1.071–3.077)	0.0266	1.886 (1.117–3.185)	0.0176	1.817 (1.090–3.028)	0.0219	1.755 (1.014–3.039)	0.0445
Tumor size	1.231 (0.776–1.953)	0.3765	1.645 (0.955–2.835)	0.0730	1.474 (0.866–2.510)	0.1526	1.411 (0.849–2.347)	0.1843	1.438 (0.848–2.439)	0.1774
Recurrence	0.924 (0.583–1.465)	0.7368	0.848 (0.516–1.392)	0.5144	0.900 (0.552–1.465)	0.6709	1.043 (0.648–1.679)	0.8619	0.782 (0.470–1.304)	0.3464
IDH1 mutant	0.356 (0.191–0.662)	0.0011	0.517 (0.270–0.993)	0.0474	0.455 (0.239–0.866)	0.0165	0.412 (0.216–0.782)	0.0067	0.533 (0.279–1.019)	0.0569

*p*-value

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
