# Peer review of "Increased Vascular Adhesion Protein 1 (VAP-1) Levels Are Associated with Alternative M2 Macrophage Activation and Poor Prognosis for Human Gliomas"

_diagnostics, 2020, doi:10.3390/diagnostics10050256_

Round 1
Reviewer 1 Report
Need to discuss the markers that were used to identify total TAMs vs M2 TAMs in the introduction. There is a lot of people that do not know about M1-M2 phenotypes, and this is not really explained.
Do not understand the premise "VAP-1 with targeted M2 immunity might be an effective therapeutic strategy for human glioma"
In this study this is not really addressed, you only showed that in more aggressive tumors there is an increase in M2 positive TAMs and this coincides with VAP-1 expression. But the mechanisms that might result in this increase, and whether this is just a coincidence or a real therapeutic opportunity is not really address. Therefore that statement is not accurate. We can only use the information provided in this paper as a prognostic indicator.
Another thing that was not explored and would be potentially important is how much of the TAMs in the gliomas represented macrophages vs microglia. Although this cells have functional similarities, they are not the same and there might be differences depending on their proportions.
Author Response
We thank you for your insightful comment, which may help reinforce the concept that we attempted to explore in this manuscript. Please refer to our following reply.
1.
Need to discuss the markers that were used to identify total TAMs vs M2 TAMs in the
introduction. There is a lot of people that do not know about M1 M2 phenotypes, and this is
not really explained.
(line 8
7 92 M1 macrophages induce inducible nitric oxide synthase (iNOS) that catalyzes the
oxidation of arginine to produce a great deal of nitric oxide (NO) as a defense mechanism.
NO causes not only cytotoxicity but also generation of numerous toxic metabolites that
c ompose the M1 killing machinery iNOS is reported to increase M1 polarization and acts as a
signature marker for M1 mediated immune responses [
(line 9
4 9 8 The majority of TAM subpopulations in tumor milieus exhibit an M2 like
phenotype exerting im mune tolerance and protumor effects [16,17]. Overall TAM subtypes
were identified by CD68, a macrophage lineage marker [16,18]. M2 TAMs are specifically
characterized with enhanced CD163 expression, attributable to the high expression of
macrophage scaveng er receptors by the M2 TAMs [17,19 22].
2.
Do not understand the premise "VAP 1 with targeted M2 immunity might be an effective
therapeutic strategy for human glioma" glioma". In this study this is not really addressed, you only
showed that in more aggressive tumors there is an increase in M2 positive TAMs and this
coincides with VAP 1 expression. But the mechanisms that might result in this increase, and
whether this is just a coincidence or a real therapeutic opportunity is not really address.
Therefore that statement is not accurate. We can only use the information provided in this
paper as a prognostic indicator.
(line 5
3 5 9 ) In conclusion, VAP 1 and TAM CD163 M2 coexpressio n was found in glioma
tissues belonging to a highly malignant subgroup that was associated with poor prognosis.
These results implied VAP 1 abundance is closely linked to alternative M2 activation during
glioma progression. From the aforementioned data, a reasonable inference is that VAP 1
combined with targeting M2 immunity might be an effective therapeutic target for human
gliomas.
3.
Another thing that was not explored and would be potentially important is how much of
the TAMs in the gliomas represented macrophages vs microglia. Although this cells have
functional similarities, they are not the same and there might be differences depending on
their proportions.
W
e agree the reviewer's comment about the differences of macrophage and microglia, but we
didn t use the specific biomarkers to identifying two types of phagocytic cells in c urrent study .
Therefore, we could not clearly define the ratio of macrophages vs microglia involving in the
TAMs within gliomas. We’ll focus on this interesting point in the further.
Reviewer 2 Report
In the manuscript by Chang et al., the authors reported that the expression of AOC3 gene (corded VAP-1 protein) and the methylation status of AOC3 gene in glioma sample are associated with the disease severity, as well as the prognosis in glioma patients from TCGA glioma and glioblastoma database set. In addition to the database analysis, the authors demonstrated that VAP-1 expression is correlated with the expression of the TAM marker, particularly that of CD163, in paraffin specimens of glioma patients. The histological examination also revealed that co-expression of VAP-1 and CD163 is strongly correlated with clinical signatures of glioma patients, particularly with the severity, as well as poor outcome of glioma patients. Thus, VAP-1 can be a favorable marker for an advanced subgroup of glioma patients associated with poor prognosis, especially when it is co-expressed with M2 marker CD163, in glioma specimen.
Overall, the data are well presented, and the manuscript is clearly written by the authors. However, there are some minor issues that should be addressed for the publication in Diagnostics.
Specific comments
- The authors should precisely describe their results in the abstract. For instance, the authors described “The bioinformatic analysis….VAP-1 (AOC3) was favorable for predicting glioma progression and patient survival” in the abstract section (Page1, line 38-39). We cannot assume whether VAP-1 expression is associated with good or poor prognosis of glioma patients. In addition, the sentence “VAP-1+ was expressed……a significant positive association with overall survival in Kaplan-Meier analysis (p<0.0001)” (Page1, line 41-43) should be corrected to “ ….a significant association with poor prognosis in …..”
- The word “in silico analysis” is inappropriate usage for such analysis because the database consists of genetic and clinical information from patients.
- It is very difficult to read the text written in Fig1 A-C. The authors should describe it with a larger font size.
- In Figure 1B and 1C, the order of low-grade glioma and glioblastoma datasets should be replaced with each other, likewise in Figure 1D and 1E.
- In Figure 1F and 1E, the authors compared the overall survival of glioma patients between AOC3+ and AOC3- subgroups, and AOC3 methylation+ and AOC3 methylation- subgroups, respectively. However, the definition of each cohort and the number of samples in each cohort are missing in the manuscript. The authors should describe such information.
- In Figure 2G-2I, the authors should describe the number of patients in each subgroup.
- In Figure S1, the authors demonstrated that CD163 expression is also strongly correlated with poor OS, indicating that CD163 expression alone is also a potential biomarker for diagnosis of glioma patients. Therefore, the ROC curve analysis of CD163 expression alone would be important to show additionally.
- The authors discussed the role of VAP-1 and CD163+ M2 macrophages in an adjacent location of glioma tissues, and suggested that VAP-1 modulates M2 macrophage infiltration. Given that M2 macrophages play an important role in the vascular formation, the authors should also discuss the possibility, in which infiltrated M2 macrophages enhances the expression of VAP-1 through the promotion of angiogenesis in glioma.
Author Response
We thank you for your insightful comment, which may help reinforce the concept that we attempted to explore in this manuscript. Please refer to our following reply.
1. The authors should precisely describe their results in the abstract. For instance, the authors described “The bioinformatic analysis….VAP-1 (AOC3) was favorable for predicting glioma progression and patient survival” in the abstract section (Page1, line 38-39). We cannot assume whether VAP-1 expression is associated with good or poor prognosis of glioma patients.
(line 38-40) Bioinformatic analysis confirmed that VAP-1 (AOC3) expression is associated with poor prognosis of glioma patients (p=0.0283).
2. In addition, the sentence “VAP-1+ was expressed……a significant positive association with overall survival in Kaplan-Meier analysis (p<0.0001)” (Page1, line 41-43) should be corrected to “ ….a significant association with poor prognosis in …..”
(line 43) “…and this phenotype revealed a significant association with overall survival in..”
3. The word “in silico analysis” is inappropriate usage for such analysis because the database consists of genetic and clinical information from patients.
(line 36) “An in depth analysis of AOC3 gene….”
4.
It is very difficult to read the text written in Fig1 A C. The authors should describe
it with a larger fo nt size.
(line 241-244) Analyses of AOC3 exon expression and DNA methylation in glioma subgroups were conducted using UCSC Xena (Figure 1B and 1C). Three exons were used in AOC3 transcriptional dataset, and twelve methylation probes were utilized in the gene methylation profile study as described in the Method Section. (line 247-250) Further, the association between AOC3 gene expression and clinical characteristics in glioma subgroups was observed by Chi-square test and performed by box-whisker plots. The patients were divided into low (-) and high (+) expression groups based on the cut-off of the median of the AOC3 exon expression (0.4471) and methylation status (0.7207).
5.
In Figure 1B and 1C, the order of low grade glioma and glioblastoma datasets
should be replaced with each other, likewise in Figure 1D and 1E.
Fig1B and 1C, these analyses of AOC3 exon expression and DNA methylation in subgroups of low-grade glioma and glioblastoma were was conducted using UCSC Xena (https://xenabrowser.net). Therefore, the order of LGG and GBM datasets was
limited by the presenting way of the analysis system.
6.
6. In Figure 1F and 1E, the authorsIn Figure 1F and 1E, the authors compared the overall survival of glioma patients compared the overall survival of glioma patients between AOC3+ and AOC3between AOC3+ and AOC3-- subgroups, and AOC3 methylation+ and AOC3 subgroups, and AOC3 methylation+ and AOC3 methylationmethylation-- subgroups, respectively. However, the definition of each cohort and the subgroups, respectively. However, the definition of each cohort and the number of samples in each cohort are missing in the mannumber of samples in each cohort are missing in the manuscript. The authors should uscript. The authors should describe such information.describe such information.
(line 247-250) The patients were divided into low (-) and high (+) expression groups based on the cut-off of the median of the AOC3 exon expression (0.4471) and methylation status (0.7207).
7.
7. In In Figure 2GFigure 2G--2I, the authors should describe the number of patients in each 2I, the authors should describe the number of patients in each subgroup.subgroup.
The numbers of patients in each subgroup were showed in Figure 2G-2I.
8.
8. In Figure S1, the authors demonstrated that CD163 expression is also strongly In Figure S1, the authors demonstrated that CD163 expression is also strongly correlated with poor Ocorrelated with poor OS, indicating that CD163 expression alone is also a potential S, indicating that CD163 expression alone is also a potential biomarker for diagnosis of glioma patients. Therefore, the ROC curve analysis of biomarker for diagnosis of glioma patients. Therefore, the ROC curve analysis of CD163 expression alone would be important to show additionally.CD163 expression alone would be important to show additionally.
The ROC curve analyses of TAM biomarkers expression alone were showed additionally in Figure S2. (line 407-408) CD163 expression alone is also a potential biomarker for diagnosis of glioma patients (AUC=0.7565, Figure S2C).
9.
9. The authors discussed the role of VAPThe authors discussed the role of VAP--1 and CD163+ M2 macrophages in an 1 and CD163+ M2 macrophages in an adjacent location of glioma tissues, and suggested that VAPadjacent location of glioma tissues, and suggested that VAP--1 modulates M2 1 modulates M2 macrophage infiltration. Given that M2 macrophages play an important role in the macrophage infiltration. Given that M2 macrophages play an important role in the vascular formation, the authors should also discuss the possibility, in which vascular formation, the authors should also discuss the possibility, in which infiltrated M2 macrophinfiltrated M2 macrophages enhances the expression of VAPages enhances the expression of VAP--1 through the 1 through the promotion of angiogenesis in glioma.promotion of angiogenesis in glioma.
(line 480-484) Several studies demonstrated that M2 TAMs display proangiogenic properties by secreting a variety of angiogenic factors and cytokines, while engaging in a number of heterotypic interactions [51-55]. VAP-1 selectively enhances M2-like TAM recruitment depending on the implanted cytokines [24, 31], and might thus indirectly contribute to M2 macrophage–mediated angiogenesis in tumors [36].